


# Storm surges and storm wind waves in the Caspian Sea in the present and future climate

Anna Pavlova[1,2], Stanislav Myslenkov[1,2,3], Victor Arkhipkin[1], Galina Surkova[1]

[1] Lomonosov Moscow State University, Faculty of Geography, Moscow, Russia

[2] Hydrometeorological Research Center of the Russian Federation, Moscow, Russia

[3] Shirshov Institute of Oceanology, Russian Academy of Sciences, Moscow, Russia

*Correspondence to*: Stanislav Myslenkov (stasocean@gmail.com)

**Abstract.** This study is devoted to the analysis of the storm surges and wind waves in the Caspian Sea for the period from 1979 to 2017-2020. The models used are the circulation model ADCIRC and the wave model WAVEWATCH III with wind and pressure forcing from the NCEP/CFSR reanalysis. The modeling is performed on the unstructured grid with spacing to 300-700 m in the coastal zone. Mean and extreme values of surges, wave parameters, and storm activity are provided.

The maximum significant wave height for the whole period amounts to 8.2 m. The average long-term SWH does not exceed 1.1 m. No significant trend in the storm activity was found.

The maximum surges height amounts to 2.7 m. Analysis of the interannual variability of the surges occurrence showed that 7-10 surges with a height of more than 1 meter were obtained per year and the total duration all these surges was 20-30 days per year.

Assessment of the risk of coastal flooding was carried out by calculating the extreme values of the Sea for different return periods 5, 10, 25, 50, and 100 years. The extreme sea level values in the northern part of the Caspian Sea for the return period 100 years is close to 3 m and the areas with big surges are located along the eastern and western coasts.

Based on climatic scenarios of CMIP5, a forecast is made for the recurrence of storm wind waves in the 21st century. A statistically significant increase of storm waves recurrence in future was found, but it is not dramatically growing.

Keywords: storm surge, wind waves, ADCIRC, WAVEWATCH III, Caspian Sea, wave climate, unstructured grid, climate change

## 1 Introduction

The investigation of specific features of surges and waves in oceans and seas is an important problem. Data on mean and extreme characteristics of level and waves are needed for navigation, marine works, construction of offshore and coastal facilities, exploration and extraction of minerals, coast protection, etc.

Currently, the basic method for studying level and wave parameters in oceans and seas is the mathematical modeling. This is caused by the deficiency of instrumental observations data with the high spatial resolution and rather a long series are most often absent.

The Caspian Sea is a fully enclosed inland water body with a maximum depth to 1025 m. The northern part of the Caspian Sea is shallow (the depth is not more than 20 m), hence, the wave development is limited there. It also means that almost the entire coast is subject to the influence of non-periodic anemobaric-surges oscillation level. In the deeper central and southern parts of the Caspian Sea, a significant role in the wave development is played by the meridional stretching of the basin; therefore, the wave development at the western and eastern wind directions is limited by a short fetch (Baidin 1986). Caspian is the inconstancy of its level, the amplitude of which varied within 4 meters during the period of instrumental observations (Bolgov et al., 2007). Most researchers explain this phenomenon by changes in climatic conditions on which overlap anthropogenic factors.



Surges are mainly generated by the action of the tangential wind stress, but also to a lesser extent as a result of changes in atmospheric pressure. In this work, the term setup is used to indicate rises of water. A fall of water below the normal level is called setdown.

Earlier studies focused on statistical analysis and modeling of surges along the Volga river (Skriptunov 1958; Gershtansky 1973). However, in recent decades, the entire territory of the Caspian Sea has been covered. In the Laboratory of Marine Applied Research of the Hydrometeorological Center of Russia, has been developed and operates within the framework of operational technology a method of hydrodynamic prediction of sea level and currents in the Caspian Sea (Zil'bershtein et al., 2001, Verbitskaya et al., 2002, Nesterov et al., 2018).

The most recent studies of storm surges in the different water areas are based on the circulation models and wave models, which are used as a coupled version or separately (Bhaskaran et al., 2013; Wang et al., 2021). The unstructured computational grids provide a good result for high-resolution modeling in shallow coastal areas [Dietrich et al., 2012; Federico et al., 2021]. Therefore, we decided to use the ADCIRC and unstructured grid for the Caspian Sea.

Some papers present the results of wind wave studies for the Caspian Sea (Ambrosimov and Ambrosimov 2008; Lopatuhin et al. 2003; Gippius et al. 2016; Yaitskaya 2017; Gholamreza et al. 2017). However, the wave climate is insufficiently researched. Among the papers dealing with the analysis of long series of wave hindcast, the paper (Lopatuhin et al., 2003) should be noted, where the detailed tables of the repeatability of waves height depending on wind direction and season are plotted using the WAVEWATCH III model and NCEP/NCAR reanalysis data. In this case, the grid spacing corresponded to input data on the wind ($\sim 1.8°$). Paper (Yaitskaya, 2017) demonstrated wave climate changes in the second half of the 20th century–the beginning of the 21st century and also used the NCEP/NCAR reanalysis with the resolution of 2.5° and the SWAN wave model to simulate the wave climate. The authors of (Golshani et al., 2005) used the MIKE 21 SW model and ERA-40 reanalysis with the spacing of $\sim1.25°$. The authors of that paper modified the coefficients of whitecapping and wave breaking based on the comparison of simulation and observational data that reduced modeling errors for specific points and for a certain time period.

Kudryavtseva et al. (2016) showed the main features of the wave climate in the Caspian Sea for the period 2002–2013 based on altimetry data (Kudryavtseva et al., 2016).

It is known that the results of analysis and forecasting of waves cardinally depend on the quality of input wind data. Unfortunately, wind errors from the reanalysis are unevenly distributed in space and in time that does not allow using the fixed coefficients to increase the accuracy of wave simulation (Van Vledder and Adem, 2015).

Data of NCEP/NCAR reanalysis considerably underestimate wind speed that is proved by the results presented in (Efimov et al. 2004; Medvedeva et al. 2015; Van Vledder and Adem, 2015). The low correlation coefficient (0.7) obtained during the assessment of the skill of wave simulation based on this reanalysis was also confirmed for the Caspian Sea (Yaitskaya 2017). Some papers (Van Vledder and Adem, 2015; Myslenkov et al., 2015; Medvedeva et al., 2016) demonstrate that the use of the data of the new-generation reanalysis (for example, NCEP/CFSR or ERA-Interim) leads to the considerable improvement of wave simulation quality. Thus, the earlier data on the Caspian Sea wave climate based on the low-resolution reanalyses should be specified. The most modern analysis of waves in the Caspian Sea is presented in (Bruneau and Toumi, 2016), where the authors used the ensemble of models (ROMS, WRF, SWAN). However, the authors of (Bruneau and Toumi, 2016) provided a hindcast only for three years, and such length of the series is insufficient for the analysis of mean and extreme characteristics.

Some publications are dealing with the hindcast and climate projections of the frequency of occurrence of synoptic conditions which cause severe hydrometeorological events including storm waves (Kislov et al., 2016; Surkova et al., 2013). However, these papers also used wind data from the NCEP/NCAR reanalysis. The papers (Rusu and Onea, 2013) provide the wave energy analysis in the Caspian Sea.





The papers (Ivkina and Galaeva, 2017; Strukov et al., 2013; Zamani et al., 2009) describing the systems of operational
analysis and forecasting of waves in the Caspian Sea should be noted. These papers present simulation skill scores; however,
the analysis of mean wave characteristics for a long time period is absent.
The task of forecasting storm waves is extremely important since their fluctuations lead to changes in marine
ecosystems, the operating conditions of coastal facilities, and shipping. Not only the short-term forecast for the coming hours
and days is important, but also the climate. This is the basis for developing a climate change adaptation and mitigation strategy
(IPCC, 2012; IPCC, 2019).
Atmospheric circulation is one of the main factors forming wind field structure and its intensity. In this work, patterns
of atmospheric circulation are revealed and classified for storm cases in the Caspian Sea using the seal level pressure data for
the present climate and climate projection.
The main objective of the present paper is to determine the mean and extreme parameters of wind waves and storm
surges in the Caspian Sea. Calculations were performed using the circulation model ADCIRC and the WAVEWATCH III
spectral wave model and data of the NCEP/CFSR/CFSv2 reanalysis.

**2 Data and methods**
**2.1 ADCIRC model setup**
In order to estimate interannual and seasonal changes of wind-surges fluctuations in the Caspian Sea, the ADvanced
CIRCulation model was used (Luettich et al. 1992; Luettich and Westerink 2004). ADCIRC is a numerical model for
calculating water circulation and sea level fluctuations that solve the complete equations of motion for a moving fluid on a
rotating earth.
The model equations have been formulated using the hydrostatic pressure and Boussinesq approximations. It uses a
finite element method for discretization by spatial variables, which allows the use the unstructured grids. The approximation
in time is carried out by the finite difference method. It takes into account such parameters as Coriolis force, atmospheric
pressure, wind stress, tidal potential, and bottom friction. In the model, you can set the properties of the underlying surface.
ADCIRC also includes flooding and drying of low-lying areas, as well as river flow.
There are two options for using the ADCIRC model: as a two-dimensional depth-integrated model (2DDI), and as a
three-dimensional model (3D). In this work, we used a two-dimensional model. Exceeding the level is obtained by solving the
depth-integrated continuity equation in the Generalized Wave-Continuity Equation form (GWCE) (Westerink et al. 1994).
Velocity obtained from the solutions of momentum equations. All non-linear terms are preserved. The ADCIRC architecture
successfully allows to use this model complex in parallel computing. ADCIRC can be run with either a Cartesian or a spherical
coordinate system. The initial equations of the 2DDI ADCIRC model are:

$$\frac{\partial U}{\partial t} + U\frac{\partial U}{\partial x} + V\frac{\partial U}{\partial y} - fV = -g\frac{\partial}{\partial x}\left[\eta + \frac{P_a}{g\rho_0}\right] + \frac{\tau_{sx} - \tau_{bx}}{\rho_0 H} + \frac{M_x - D_x}{H} \tag{1}$$

$$\frac{\partial V}{\partial t} + U\frac{\partial V}{\partial x} + V\frac{\partial V}{\partial y} - fU = -g\frac{\partial}{\partial y}\left[\eta + \frac{P_a}{g\rho_0}\right] + \frac{\tau_{sy} - \tau_{by}}{\rho_0 H} + \frac{M_y - D_y}{H} \tag{2}$$

$$\frac{\partial^2 \eta}{\partial^2 t} + \tau_0\frac{\partial \eta}{\partial t} + \frac{\partial J_x}{\partial x} + \frac{\partial J_y}{\partial y} = 0 \tag{3}$$

$$J_x = -Q_x\frac{\partial U}{\partial x} - Q_y\frac{\partial U}{\partial y} + fQ_y - \frac{g}{2}\frac{\partial \eta^2}{\partial x} - \frac{H}{\rho_0}\frac{\partial P_a}{\partial x} + \frac{\tau_{sx} - \tau_{bx}}{\rho_0} + (M_x - D_x) + \tau_0 Q_x + U\frac{\partial \eta}{\partial t} - gH\frac{\partial \eta}{\partial x} \tag{4}$$

$$J_y = -Q_x\frac{\partial V}{\partial x} - Q_y\frac{\partial V}{\partial y} - fQ_x - \frac{g}{2}\frac{\partial \eta^2}{\partial y} - \frac{H}{\rho_0}\frac{\partial P_a}{\partial y} + \frac{\tau_{sy} - \tau_{by}}{\rho_0} + (M_y - D_y) + \tau_0 Q_y + V\frac{\partial \eta}{\partial t} - gH\frac{\partial \eta}{\partial y}$$

$$\tau_{sx} = \tau_{sx,wind} + \tau_{sx,wave}, \tau_{sy} = \tau_{sy,wind} + \tau_{sy,wave}$$

$$\tag{5}$$





(6)


where $x$, $y$ and $t$ are horizontal grid points and time; $H = h + \eta$ is total water depth; $\eta$ is surface elevation; $h$ is
bathymetric depth; $U$ and $V$ are depth-integration currents in the $x$- and $y$-directions, respectively; $Q_x = UH$ and $Q_y = VH$ are
fluxes per unit width; $f$ is the Coriolis parameter; $g$ is gravitational acceleration; $P_a$ is atmospheric pressure at the surface; $\rho_0$ is
the reference density of water; ($\tau_{sx,wind}$, $\tau_{sy,wind}$) and ($\tau_{sx,wave}$, $\tau_{sy,wave}$) are surface stresses due to winds and waves, respectively; $\tau_{bx}$
and $\tau_{by}$ are bottom stresses; $M_x$ and $M_y$ are horizontal eddy viscosity; $D_x$ and $D_y$ are momentum dispersion terms; $\tau_0$ is a
numerical parameter that optimizes the phase propagation properties.
Calculations were performed on the unstructured grid based on the bathymetric data of detailed navigation maps of
the Department of Navigation and Oceanography (https://navysoft.ru/). On this maps the depth was presented relative to the
sea level of –28 m (Baltic Level System), taking into account flooding and drying of the coastal land.  For the ADCIRC model
the grid consisted of 71523 points (Fig. 1). The grid spacing varied from 10 km in the open sea to 500 m in the coastal zone
(Fig. 1). The computational grid was created using the Aquaveo Surface-water Modeling System (SMS 11.0) application. The
similar computational grids used for the wave and surge modeling in the other seas of the Russian Federation proved their
efficiency (Ivanova et al., 2015; Medvedeva et al. 2016; Myslenkov et al. 2017).
As the input data were used the fields of surface wind (at a height of 10 meters) and the atmospheric pressure of the
NCEP (National Centers for Environmental Prediction) CFSR (Climate Forecast System Reanalysis) reanalysis. The
NCEP/CFSR reanalysis is a modern product of the National Centers for Environmental Prediction, implemented in 2010. The
CFSR is a global atmosphere–ocean–land–sea ice system with high resolution that provides the best estimate of the state of
these interconnected systems. Data covering the period from 1979 to 2010 have a 1-hour time interval and a spatial resolution
of $\sim 0.3° \times 0.3°$ (Saha et al., 2010).
For numerical calculations from 2011 to 2017, the NCEP/CFSv2 version (Climate Forecast System Version 2)
reanalysis was used, having a spatial resolution of $\sim 0.2° \times 0.2°$ (Saha et al., 2014). Also, sea ice concentration was set as input
(OSI-450), which is reprocessing the brightness temperature data based on passive microwave radiometer SMMR, SSM/I, and
SSMIS and ECMWF ERA-Interim reanalysis (EUMETSAT).
A separate experiment was carried out using coupled model version ADCIRC+SWAN. We used the following
configuration for SWAN model: GEN3, KOMEN (cds2 = 2.36$e$ − 5, stpm = 3.02$e$ − 3), Quadrupl, Triad, Breaking constant
(alfa = 1.0, gamma = 0.73) and Friction Jonswap constant (cf = 0.067). The spectral resolution of the model is 36 directions
($\Delta\theta = 10°$), the frequency range includes 36 intervals (from 0.03 to 1.1 Hz). The general time step for the integration of the
full wave equation was 15 minutes.



**2.2 Wave model** WAVEWATCH III setup
The WAVEWATCH III (WW3) version 6.07 third-generation spectral wave model was used to calculate wind wave
parameters for the Caspian Sea (Tolman 2014). It is known that the energy inflow to waves is provided by wind energy, and
its dissipation is caused by the breaking of wave crests due to the bottom friction or wave breaking at the critical depth.
The model is based on the numerical solution for the equation of spectral wave energy balance:

$$\frac{\partial E(\omega, \theta, \vec{x}, t)}{\partial t} + \vec{V}(\omega, \theta)\nabla E = S(\omega, \theta, \vec{x}, t),$$     (7)
where $\omega$ and $\theta$ are the frequency and direction of propagation of the spectral component of wave energy;
$E(\omega, \theta, \vec{x}, t)$ is the two-dimensional energy spectrum at the point with the vector coordinate $\vec{x}$ at time moment $t$;





$\vec{V}(\omega, \theta)$ is the group velocity of spectral components; $S(\omega, \theta, \bar{x}, t)$ is the function describing sources and sinks of wave
energy. The energy balance equation is integrated using finite-difference schemes by the geographic grid and the spectrum of
wave parameters.
The authors of the present paper used the ST6 scheme for the wave generation. The nonlinear interactions were
calculated using the DIA (Discrete Interaction Approximation) parameterization being a standard approximation for the
computation of nonlinear interactions in all modern wave models.
The influence of the sea ice on the wave development was considered by the IC0 scheme, where a grid point is
considered as ice-covered if ice concentration is >0.5.
In the shallow coastal zone, the increase in wave height as waves approach the shore and the related wave breaking
after waves reach the critical value of the steepness is taken into consideration besides the wave breaking due to the long wind
effect on the sea surface (it is taken into account in the ST6 scheme). The standard JONSWAP scheme was used to take the
bottom friction into account. The spectral resolution of the model is 36 directions ($\Delta\theta = 10°$), the frequency range includes 36
intervals (from 0.03 to 0.83 Hz).
Calculations were performed on the unstructured grid which consisted of 15792 points. The grid spacing varied from
10 km in the open sea to 900 m in the coastal zone (Fig. 2).


The general time step for the integration of the full wave equation is 30 minutes, the time step for the integration of
functions of sources and sinks of wave energy is 30 s, the time step for the spectral energy transfer and for satisfying the
Courant–Friedrichs–Lewy condition is 900 s.
The wind and ice data were the same as used for ADCIRC model (NCEP/CFSR/CFSv2 and OSI-450).
In the recent 40 years, the Caspian Sea level has considerably varied (Nesterov 2016). Therefore, different levels were
specified in the model for every year to provide more correct calculations. Year average of Caspian Sea level was calculated
by using the data from several gauging stations (Makhachkala, Tyulenii, Peshnoi, and Baku) which collected on
http://www.caspcom.com/.
A similar implementation of the WAVEWATCH III model was successfully used by the authors for studying wave
parameters in the other Russian Seas (Myslenkov et al., 2018; Myslenkov et al., 2021).
As a model output, we got the wind wave fields for every three hours from 1979 to 2020 (total 42 years). We tested
the data with a time step of 1 hour and 3 hours and did not reveal a significant change in the extremes (no more than 0.1 m).
The model results include the SWH ($4\sqrt{m_0}$, where m0 is the zero-order moment of the wave spectrum, approximately SWH
is the mean value from 1/3 of the highest waves), the wave propagation direction, the mean wave period (WP) Tm02=$(2\pi\sqrt{\overline{\sigma^2}})$,
and mean wavelength (WL)= $(2\pi\overline{k^{-1}})$. These data were used to compute maximum and average long-term values as well as
extreme characteristics.
The storm activity analysis was held according to the Peak Over Threshold (POT) method, which is widely used (De
Leo et al., 2020; Myslenkov et al., 2021). The essence of the POT method is to find an extreme values of some sample that
exceeds a certain threshold value. We used POT previously for the Barents and Kara Seas wave analysis (Myslenkov et al.,
2021). The number of storm waves with different SWH from 2 to 5 m was calculated for each year in the whole Caspian Sea.
The calculation procedure included the following steps: if at least one node in the investigated sea area had the SWH exceeding
for example a 2 m (or a different threshold from 2 to 5 m), then such event was attributed to the storm case with SWH threshold
2 m. This event continued until the SWH was not less than the threshold at all nodes of the investigated area. Further, if the
SWH threshold exceeded in one of the nodes again, then this event was added to the following case. A period of at least 9
hours passed between two storm cases for eliminating the possible errors. This technique has an in-accuracy associated with





storms running in a row or from different directions at the same time. However, such cases are rare. The proposed algorithm
works correctly; it was validated by a visual analysis conducted for several years.

**2.4 Future Climate data and methods**
Mathematical models are effective tools for predicting the state of the climate system. However, even using the most
advanced climate models, only indicators that are based on average values of climatic variables related to large territories or
accumulated amounts are confidently predicted (with sufficient accuracy for practical use). Specific problems arise in
predicting extreme events. In this paper, one of them is considered – the storm waves. A necessary (but not sufficient) condition
for the development of the storm is a high wind speed, but its direct model forecast has a lower quality than the forecast of the
atmospheric pressure field that forms the surface wind field.
Climate projection of weather hazards needs their classification. There are two fundamental approaches to
investigating the link between the large-scale circulation and environmental variables (Cannon et al., 2002; Demuzere, 2011;
Yarnal, 1993). In the framework of the first one, so-called the "circulation – to environment" approach (Demuzere, 2011),
arrangement of the circulation data of interest (e.g. sea level pressure, geopotential height, etc.) is carried out to group them
into circulation types (CTs) according to a selected methodology (clustering, principal component analysis (PCA), regression,
etc.). Then, one looks for relations of CTs with the local-scale environmental variable (e.g. storm waves). On the contrary, the
"environment – to circulation" approach is based on the classification of the circulation data for certain criteria of the
environmental variable, so that composite maps of the circulation variable can be derived for a specific environmental
condition. Both approaches are widely used in various fields of atmospheric sciences. The first one was successfully
implemented in the framework of the COST733 action (http://cost733.met.no) entitled "The harmonization and application of
weather types classifications for European regions", an extensive review of existing classification methods and those used in
COST733 is performed by Huth et al. (2008). In this study, we used the second "environment – to circulation" approach for
storm events in the Caspian Sea.
The classifications of atmospheric circulation are very useful tools in climate change research; for the reconstruction
of the past climate, analysis of variability of the present climate, and in the estimates of future climate. The practical application
is to produce climate projection and for a more specific purposes, for example, for storm waves frequency in the future, as it
was done in this study.
The wave height of 3 m and more is chosen as a criterion of the storm day. According to  this data the storm calendar
was created. For these days sea level pressure (SLP) fields are used to classify circulation types. Sea level pressure daily data
was taken from reanalysis NCEP/CFSR (Saha et al., 2010; 2014) for 1979-2017. Then, on the base of SLP fields, atmospheric
circulation patterns were obtained by cluster analysis (k-means approach (e.g., Hartigan, Wong, 1978) preprocessed by
Empirical Orthogonal Function (EOF) analysis (e.g., Preisendorfer, 1988) to reveal few leading modes determining the most
part of the variance. These techniques of EOF decomposition and k -means cluster analysis together or in combinations with
other techniques are widely used in circulation types classifications (Corte-Real et al., 1999; Cannon et al., 2002; Solman,
2003; Stahl, 2006; Cassou, 2010; Philipp et al., 2010; Santos et al., 2011, etc.).
At the first stage of classification, for every storm case from the calendar, it was prepared a dataset consisting of 30
daily Sea Level Pressure (SLP) grids – 15 days before and after storm day. After EOF decomposition of daily SLP grids, the
first three eigenvectors explaining more than 70% of the variability were retained (Surkova et al., 2013), thus, filtering high-
frequency perturbations (SLP-EOF fields).
SLP-EOF fields for storm days (according to the storm calendar of the sea) were used as input variables to classify
circulation patterns. Definition of circulation types was carried out using the k-mean cluster analysis. The k -means algorithm
starts with a preset number of clusters k and then moves objects between clusters with the goal of, first, minimizing the variance
within clusters and, second, maximizing the variance between clusters. Cluster centroids (ensemble mean of cluster members)




were constructed for each circulation type by averaging the SLP grids of all days that belonged to the same circulation type.
The dependence of assigning SLP field to a certain cluster on the area size was checked: we started from initial territory 0–
90E, 30–80N and reduced it gradually until it was comparable with the sea size. It almost did not influence the result of sorting.
Finally, we left area 0–90 E, 30–80 N for centroids construction to have a full large-scale synoptic view of the circulation
pattern.
Then daily SLP data of CSFR Reanalysis for the period 1979–2017 was sorted considering previously derived
circulation patterns. As one of the distance measures (Huth et al., 2008; Brinkmann, 2000; Lund, 1963), space correlation was
used between model data and reanalysis SLP-EOF fields for days from the storm calendar. To eliminate 'noise' on the
boundaries of a rather large initial domain (0–90E, 30–80N) but to save individual features of the SLP field, a correlation was
estimated for 30–70E, 30–70N for the Caspian Sea.  The spatial scale of the smaller domains is comparable with the size of
such typical mid-latitude synoptic structures as cyclones and anticyclones governing surface winds and therefore storm waves.
Next, the spatial correlation was calculated for storm days and CMIP5 models for Historical climate experiments and
for RCP8.5 scenario of the future climate (Moss et al., 2010). We used data of the following models: ACCESS1.0, bcc-csm1-
1, BNU-ESM, CanESM2, CCSM4, CESM1-BGC, CESM1-CAM5, CMCC-CESM, CMCC-CMS, CNRM-CM5, CSIRO-
Mk3.6.0, GFDL-CM3, GFDL-ESM2G, GISS-2-H, GISS-E2-R, HadGEM2-CC, HadGEM2-ES, INMCM4, IPSL-CM5A-LR,
IPSL-CM5A-MR, IPSL-CM5B-LR, MIROC-ESM, MIROC-ESM-CHEM, MIROC5, MPI-ESM-LR, MPI-ESM-MR, MRI-
CGCM3, MRI-ESM1, NorESM1-M.  Before the correlation procedure, all models data were interpolated on the reanalysis
grid.

### 3 Results and discussion

### 3.1 Storm surges

To assess the quality of the ADCIRC model, the obtained model level results were compared with the level
measurements on the weather stations Tuleniy Island and Makhachkala (with a time step of 6 hours) from 2003 to 2017. The
correlation coefficient for the Tuleniy Island varied between 0.79 – 0.88 (Fig. 3), for Makhachkala – 0.67 – 0.79. The mean
value of the root-mean-square error was about 0.11 m for the Tuleniy, and 0.06 m for the Makhachkala. If we exclude
fluctuations in sea level with an amplitude of less than 10 cm, the correlation values increase to 0.94. The quality of the
modeled data allows estimating the regular and extreme characteristics of the storm surges. A more detailed description of
statistical analysis was given in (Pavlova et al. 2020).



At the next stage, we carried out a special experiment with coupled ADCIRC + SWAN models to understand the
contribution of wind wave setup in storm surges. We have performed calculations for several of the strongest surges. Coupled
ADCIRC + SWAN experiments showed that the average contribution of wave setup to the height of the surge does not exceed
5 cm. Two examples of surges are shown in Fig. 4. In further analysis, we used the results of the single ADCIRC model
configuration and did not take into account the wind wave setup, since it is very small.


The main task of this work was to analyze the interannual and spatial distribution of storm surges for the period 1979-
2017. The distribution of the maximum surge heights for the Caspian Sea for the modeling period (1979–2017) is shown in
Figure 5. Analysis spatial distribution of the maximum surges height showed that in the central part of the Caspian Sea maximal
surges height do not exceed the 0.3 m. In the south part of the Sea, maximal surges height do not exceed the 0.8-0.9 m and is
located in the southeast. In the northern part of the Caspian Sea, there are two areas of maxima. The first area is in the





northwestern part along the coast near the Volga river, and the second in the east, where surges reached 2.5 and 2.7 m,
respectively (Fig. 5).

The depth in the northern part of the Caspian Sea is not more than 20 m and this part is semi-closed, thus it is probably

the main reason for the strong surges in this area.

The next step of our research was an analysis of the main synoptic situations which lead to the formation of storm

surges. By the term storm surge, we mean the deviation of the sea level from the average annual level by a certain amount.
This analysis is based on a model output sea level data from 1979 to 2017 in all points of the computational grid. The storm
surge was calculated using POT method. We found all cases when the surge was more than 1 m and analyzed the synoptic
situation which leads to these surges.

Three main synoptic situations have identified that lead to the formation of surges of more than 1 meter:

1) For the formation of surge on the west coast, a strong anticyclone is formed north of the Caspian Sea, which comes

either from the Asian maximum or from the north. At the same time, a cyclone is formed west of the Caspian Sea (in most
cases over European territory).

2) The opposite situation is formed for the formation of the sea level decline. That is a strong cyclone forms to the

north of the Caspian, and an anticyclone in the west.

3) The formation of a cyclone directly above the Caspian Sea, which moves to its northern part and forms a set-down

of the sea level on the west coast.

One of the important tasks in the study of storm surges is to identify the contribution of the main factors to the surge

development. To estimate the contribution of wind stress and atmospheric pressure to the formation of setup and setdown sea
level, we made numerical experiments where the circulation model is forced only by the wind or atmospheric pressure. In both
cases, ice fields were taken into account in calculations. As a result of comparing sea level according to different experiments
was obtained that wind forcing is responsible for the largest changes in water level due to surge 92 – 100 % of total surge
height and the atmospheric pressure induced 0 – 8 % in different synoptic situations. We understand that there are nonlinear
and resonant effects under the simultaneous influence of pressure and wind, however, even the analysis showed that the
contribution of wind is certainly much greater.

The number of surges events per year was calculated in the Caspian Sea according to the POT method (the technique

is described in Section 2.2 and it was the same for surges and wind wave events). The storm surges events was calculated with
different thresholds 0.5 and 1 m. Setup and setdown events are calculated and analyzed separately.

If we analyze the whole sea the average number of surges are 7-10 surges per year with a height of more than 1 meter

and a total duration of all surges in total to 20-30 days per year. Figure 6 shows an example of the distribution of the number
of surges for 1981.

For a more detailed analysis of surges number in the north part of the Caspian Sea, 6 points along the coast were

selected. The depth in those points is more than 1 meter (at sea level - 28 m BS): 1 and 2 - in the west, 3 - in the north, 4 and
5 - in the east, 6 - in the southeast (Fig. 7).

The number of surges (setup and setdown events) for each year for 6 different points is shown on Figure 8. From

1979 to 2005-2008 there is a long-term significant trend to reduce the number of surges in all points. For example, in the 1
point the number of surges with a threshold 0.5 m reduced from ~25 to 15 cases (Fig. 8). The significance of trends was
assessed by the F-test. The F-statistic is the standard test statistic for testing the statistical significance test of the linear model.
We have applied the F-test statistics of the analysis of variance (ANOVA) approach, which based on the null-hypothesis that



the means of a given set sample of normally distributed populations, all having the same standard deviation, are equal. Trends
for all points except point 2 are significant at the level p = 0.05.


After 2005-2008 the rise of the number of surges was obtained in all points (Fig. 8). If we consider the changes in
the long-term average annual Caspian Sea level fluctuations, we can see the sea level rise from 1978 to 1995, and sustained
reduction from 2005 to 2017 (Fig. 9). These fluctuations are associated with climate change (Nesterov 2016). We can see a
slight negative correlation between sea level and the number of surges. Since the surges are observed mainly in the northern
part of the Caspian Sea, they are formed by winds of the southern direction. On the other hand, with a southern wind, there is
usually less precipitation, less humidity, and more evaporation. Thus, with an increase in the recurrence of southern winds and
storm surges, we can see the reduction of sea level due to climate changes in a water balance.

The number of surges of more than 0.5 meters from 1979 to 2017 on the west coast varies from 10 to 36 and from 12
to 41 in the east (Fig.8). In the north, the values do not exceed 26. In case of surges of more than 1 meter, the maximum number
of surges of up to 15 cases per year.
The number of setdown over 1 meter is on average more than the number of setup. During 39 years the total number
of setup for the 1 point was 140, 2 point – 109, 3 point – 40, 4 point – 177, 5 point – 234, 6 point – 32. The number of setdown
for the 1 point was 241, 2 point – 144, 3 point – 109, 4 point – 245, 5 point – 303, 6 point – 9.
A more detailed examination of the intra-annual variability indicates that on the west coast maximum number of
surges occur in January and February, while on the north and east coast - in February and March. The minimum number of
surges is observed in the summer.
On the western coast, at points 1 and 2, surge heights of more than 2 m were observed 2 times during the study period.
The maximum was reached on February 17, 1981 with a value of 2.14 m, and on March 30, 2015 - 2.4 m. In the north, at point
3, the maximum surge was 1.77 m and was observed on April 10, 1979. On the east coast, at points 4 and 5, surge heights of
more than 2 m were observed 9 and 6 times, respectively. At the same time, the largest surge at point 4 was observed on
January 12, 1990, and reached 2.25 m, and at point 5 it was observed on February 1, 1981, with values of 2.71 m. In the
southeast at point 6, the maximum surge height was 1.77 m and was observed on February 3, 1981.
In this work, to assess the risk of coastal flooding, the extreme sea level values were calculated for different return
periods: 5, 10, 25, 50, and 100 years.
In accordance with the National Standard of the Russian Federation (GOST R 58112-2018), the regime distributions
of the total sea level heights belong to the type of exponential distributions, and, therefore, the Gumbel distribution is used to
determine the extreme characteristics:

$$F(V) = e^{-e^{-\frac{V-A}{B}}},$$

parameters $A, B$ can be determined from a ranked sample of $N$ annual maximum heights of the total level using the least squares
method:

$$R_i = -\ln\left[-\ln\left(1-\frac{i}{N}\right)\right], \ i = 1, N;$$

$$A = \frac{\sum_i V_i \sum_i R_i^2 - \sum_i V_i R_i \sum_i R_i}{N \sum_i R_i^2 - (\sum_i R_i)^2}, \qquad B = \frac{N \sum_i V_i R_i - \sum_i V_i \sum_i R_i}{N \sum_i R_i^2 - (\sum_i R_i)^2}.$$

After determining the parameters $A$ and $B$, $V_T$ values possible once in $T$ years, are determined as quantiles $\left(1-\frac{1}{T}\right)\cdot$
100% probability distribution defined by F(V),

$$V_T = A - B \cdot \ln\left[-\ln\left(1-\frac{1}{T}\right)\right].$$





365         V. Langbein established the relationship between the period of the frequency and number of values used in the sample

that exceeds a certain value of the statistical variable:

$T' = 1/[1 - \exp(-m'/N)]$, где $N$ - is the total number of data.

Figure 10 shows a graph of extreme sea level values for different return periods 5, 10, 25, 50, and 100 years. The

minimum extreme values are observed at points 3 and 6 in the north and southeast, and the maximum - at points 4 and 5 in the
east. This is due to the prevailing winds causing the largest surges on the western and eastern coasts, while the wave of level
rise passes along the northern and southeastern coasts. On the east coast, the level with a return period 100 years corresponds
to 2.9 m, and on the west – 2.6 m.

Figure 11 shows the extreme sea level values in the northern part of the Caspian Sea for the return periods 10 and 100

years. The maximum value for a return period 100 years is close to 3 m and the areas with big surges are located along the
eastern and western coasts.


**3.2 Extreme wind waves**

At the first stage of the wind wave climate studying, the quality of wave model results was assessed (due to the

absence of digital data of direct measurements, observational data from (Ambrosimov and Ambrosimov 2008) were used).
The results of the comparison of model and observational data for the wave heights of 3% probability of exceedance for the
point located in the Central Caspian Sea are presented in Fig. 12a. Visually, the simulation quality may be assessed as
satisfactory. The model simulates well the phase of the storm beginning; the model does not simulate storm peaks very
accurately, but no systematic underestimation or overestimation of data is observed. Hindcasts were also compared with AltiKa
altimeter data (rads.tudelft.nl). The significant wave height (SWH) at 34 990 points for the period of 2013 to 2016 was used
(Fig. 12b). The correlation coefficient was 0.918, the root-mean-square error, 0.28 m, and the BIAS, 0.07 m Scatter Index
0.29. Such skill of simulation generally corresponds to the modern realizations of wave models (Nesterov 2013; Strukov et al.
2013; Myslenkov et al.; Van Vledder and Adem 2015).

According to section 3.1 the strong storm surges are observed in the northern part of the Caspian Sea so that the sea

level can vary by 0.5–1.5 m during several days. The water depth is changing which can take effect the wave parameters. To
estimate the effect of strong surges on the wave simulation for the northern part of the Caspian Sea, a special numerical
experiment was performed. For October 1984, when the strong storm surge was registered, waves were simulated using data
on the average annual sea level, and with the sea level increased by 1.5 m. The analysis of obtained data revealed that in the
part of the northern part of the Caspian Sea where the depth is more than 7 m and waves with the maximum height (>2 m) are
observed, the depth increase does not affect the wave height. Evidently, the main factor limiting the wave growth in this area
of the Caspian Sea is a short fetch. In the areas where the sea depth is less than 2–3 m, its increase led to the wave height
growth by 5–10%. Therefore, the effect of surges on mean and extreme characteristics of waves is not considered in the present
paper.

The results of model calculations for the period of 1979 to 2020 were statistically processed. Figure 13 a, b presents

the long-term mean and maximum SWH (13% probability of exceedance) over the whole simulation period. The multiyear
mean wave height in the central part of the Caspian Sea reaches 1.1 m.  The maximum value of wave height is 8.2 m, which
is observed in the central part of the Caspian Sea. In the south part of the Caspian Sea, the local maximum SWH is equal to
6.9 m near the coast of Iran. The wave development in the northern part of the Caspian Sea is essentially limited due to the
short fetch, small depth, and ice presence; therefore, the maximum wave height is 2.7 m. Maximum values for the mean wave
period are 7.6, 7.1, and 4 s for the central, south, and north parts of the Caspian Sea, respectively. Maximum values of the
mean wavelength are typical of the central part of the Caspian Sea and amount to 163 m.





According to the data of wave climate book (Lopatoukhin et al., 2003), the maximum height of 3% probability waves
(which can be observed in the Caspian Sea once in 50 years), is 7 m. In our results, we have a maximum value of SWH 8.2 m,
which can be transformed to 10.8 m of 3% probability of exceedance. Probably, the use of NCEP/NCAR reanalysis data on
the wind by the authors of (2003) for calculations considerably underestimated the wave height. In our previous results we get
the maximal SWH about 8.9 m (Myslenkov et al., 2018), but it was old ST1 scheme for WW3 model with the same wind data.

Figures 14. *a-h* presents the maps of the distribution of the long-term mean and maximal SWH for different seasons.
For this purpose, maximum values for winter (December–February), spring (March–May), summer (June– August), and
autumn (September–November) were selected from the whole series of SWH. Maximum waves are observed in winter in the
Central Caspian Sea closer to the Absheron Peninsula. A similar situation approximately in the same region is typical of
autumn, when the wave height is 7.2 m. In summer, the height of waves with such probability does not exceed 5.7 m.
Maximum values of the long-term mean SWH is observed in winter in the Central Caspian Sea and it does not exceed
1.4 m. In summer long-term mean Maximum value of mean SWH is 0.8-0.84 m.

The number of storm events per year was calculated in the whole Caspian Sea according to the POT method (the
technique is described in Section 2.2). The events have different SWH thresholds from 2 to 5 m. Next, we will call these storm
events with a different wave height simply as a storm. At first, we analyzed the number of storms for each year (Fig. 15),
which is called the recurrence of a storm. Cases of storms with the SWH ≥ 2 m were observed about ~90-110 times per year,
with maxima in 1998 and 2009. The number of storms with the SWH ≥ 3 m is about ~40-60 times. The storms with a SWH
threshold 5 m are about 1-8 times per year. The local maximum number of storms simultaneously for different threshold SWH
≥2-4 m was in 1998. Other maxima are observed either for one threshold or for another. A local maximum number of storms
with SWH thresholds 3 and 4 m was noted in 1995. The minimum number of storms for several SWH thresholds were noted
in 2000. A linear positive trend in the number of storms is observed for almost all SWH thresholds. An increase in storm
recurrence was observed for cases with thresholds 3 m from 40 to 51 cases for all period 1979 to 2020. But all trends are weak
and insignificant.
It is worth noting that there is high interannual variability in the number of storms. The average variance is about 8%
from year to year for storms with an SWH threshold 2 m and 60% for SWH thresholds 5 m.
The most interesting feature is the significant linear trend for the storms with SWH threshold ≥ 3 m from 2003 to
2016 (Fig. 15). An increase for this period was obtained from 30 to 65 and very well consistent with an increase in the number
of surges of more than 0.5 meters which was also obtained for the period from 2005-2008 to 2016.



**3.3 Future climate projections**
For the situations of storm waves events, the typization of the surface atmospheric pressure fields was performed
using the cluster analysis method. Three main types were identified (Fig. 16). These types by genetically and circulation
features are consistent with the five synoptic types developed earlier (Madat-zade, 1954, 1959) and five types of wind fields
(Koshinskiy, 1975). However, three types identified in this work are more generalized.
The first type of circulation pattern is characterized by the development of a powerful anticyclone in the east of the
European part of Russia, the Middle Urals, and Western Siberia. Cyclonic activity is forced out to the north and to the
Mediterranean Sea. The southern periphery of the anticyclone determines the wind regime over the Caspian. Almost half of
the storm cases are attributed to the second type. Its distinguishing features are the extensive cyclonic region over the Arctic,
cyclones over the Middle East and Iran, two large anticyclones with centers over southeastern Europe and Mongolia. Between




anticyclones over Central Asia, there is a low-pressure jumper, which extends to the Caspian. The third type has some
similarities with the second type, but it is distinguished by a less pronounced anticyclone over Europe and a pronounced region
of reduced pressure over Central Asia. In this case, increased cyclonic activity contributes to the development of high wind
speeds above the sea. In percentage terms, type II and type III dominate Fig. 16).

Figure 17 shows the long-term dynamics of the anomalies of the general (accumulated) decades of repeatability of
synoptic situations, analogs of which in today's climate are accompanied by storm waves (with a SWH threshold $\geq$ 3 m). The
results obtained from the ensemble of climate models indicate that when the most unfavorable climatic scenario RCP8.5 is
implemented during the 21st century, a gradual increase in the frequency of occurrence of such situations is possible. Most of
the scenarios give an increase in synoptic situations leading to storms. In he present climate, the number of storms with the
SWH threshold $\geq$ 3 m is 40-60 cases and if we will get the positive anomaly 10-12 cases per 10 years it is not dramatically
growing.


**4 Discussions and Conclusions**
This article presents new information about storm surges, wind waves, and their recurrence in the Caspian Sea based
on the results of numerical modeling. Long-term calculations were performed using the ADCIRC and WAVEWATCH III
models.
The storm surges maximum is 2.7 m and it was observed in the northern part of the Sea for the modeling period
(1979–2017). The northern part of the Caspian Sea is shallow and semi-closed, thus it is probably the main reason for the
strong surges in this area.
The contribution of the wind stress in sea level fluctuations due the surge is 92 – 100 % and the contribution of the
atmospheric pressure is 0 – 8 % in different synoptic situations.
The number of surges events per year was calculated according to the POT method for the whole Sea or for the
different points in the northern part of the Caspian Sea. There are 7-10 surges per year (on average) with a height of more than
1 meter for the whole sea. Based on analysis of the number of surges in the different points we obtained that from 1979 to
2005-2008 there is a long-term significant trend to reduce the number of surges in all points. After 2005-2008 the rise of the
number of surges was obtained in all points. A slight negative correlation between the average annual sea level and the number
of surges obtained. The number of surges on the east coast is greater than on the west for the northern part of the sea.
The extreme sea levels for different return periods (5, 10, 25, 50, and 100 years) were calculated. The extreme sea
level values in the northern part of the Caspian Sea for the return period 100 years is close to 3 m and the areas with big surges
are located along the eastern and western coasts.
The mean SWH for the entire sea varies from 0.7 to 1.1 m. The SWH maximum is 8.2 m and it was observed in the
central part of the Caspian Sea. Analysis of the SWH maxima for different seasons showed that the SWH does not exceed 5.7
m in Summer.
The storm recurrence with the SWH $\geq$ 2 m was observed about ~90-110 times per year, with maxima in 1998 and
2009. The number of storms with the SWH $\geq$ 3 m is about ~40-60 times. The storm recurrence with SWH $\geq$5 m is about 1-8
times per year. The local maximum number of storms simultaneously for different threshold SWH $\geq$2-4 m was in 1998. A
linear positive insignificant trend in the number of storms is observed for almost all SWH thresholds.
The significant linear trend for the storm recurrence with the SWH threshold $\geq$ 3 m from 2003 to 2016 is very well
consistent with an increase in the number of surges of more than 0.5 meters which was also obtained for period since 2005-
2008 to 2016.



Synoptic patterns of storm waves (SWH ≥3 м) situations are classified for the modern climate and projection of their
frequency on the base of CMIP5 scenario RCP8.5 showed a statistically significant increase of these situations, but it is not
dramatically growing.
However, there are several discussion points in the obtained results. The first point is a quality assessment of the
circulation and wave models for the extremely high surges or waves. Unfortunately, the authors do not have full-scale direct
measurement of sea level and wave height data in the Caspian Sea. We have the maximum value of sea level 1.4 m from direct
measurements. We calculated the RMSE around 0.1 m for sea level from the ADCIRC model. What will be the quality of the
model for a surge height of more than 2-2.5 m? Unfortunately, we do not know this, and the solution to this problem could be
long-term sea level monitoring, which is currently not very active in the northern part of the Caspian Sea or the data is not in
open access.
In the case of quality assessment of the wave model we have the maximum SWH 5.5 m from direct measurements
and maximum SWH 4.8 m from satellite data. An additional problem is that the presence in measurement data of 1-2 cases of
storms with SWH 5-6 m height or a surge with the height of 1.2-1.4 m is completely insufficient for statistically acceptable
quality assessments for this range.
In our previous results, we get the maximal SWH о about 8.9 m (Myslenkov et al., 2018), but we used the old ST1
scheme for WW3 model. The quality assessment of the previous version of calculations based on the same satellite data showed
exactly the same quality of the model for the whole range of SWH. However, we found that the ST1 scheme provide
overestimates the SWH more than 4-5 m (Fig. 18) and we decide to use ST6 scheme which provides the same quality at the
whole range and slightly better quality in high waves (Fig. 12). However, we are not sure that the modeling of extreme waves
has been successful since there are very few measurement data for the high wave range.

The same goes for the calculation of surge heights, especially extreme events with return periods 50 or 100 years.
Because the distribution function is highly dependent on the values of extreme surges. If you slightly change the maximum
surge height (which is provided by long-term modeling), then the distribution function will change.

**Acknowledgments.** The surge simulation was carried out by V. S. Arkhipin and A. V. Pavlova and was supported
by the Russian Foundation for Basic Research (grant 18-05-80088). The work of S.A. Myslenkov was supported by the
Interdisciplinary Scientific and Educational School of Moscow State University "The Future of the Planet and Global
Environmental Changes."
**Data availability**
Data and results in this article resulting from numerical simulations are available upon request from the corresponding
author.
**Author contributions**
The concept of the study was jointly developed by AP and VA. AP did the numerical simulations of storm surges,
analysis, visualization and manuscript writing. VA did the numerical simulations of storm surges and analysis. SM did the
numerical simulations of wind waves and analysis of storm activity. GS did the future climate analysis. SM prepared the paper
with contributions from AP, VA, and GS.
**Competing interests**
The authors declare that they have no conflict of interest.

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





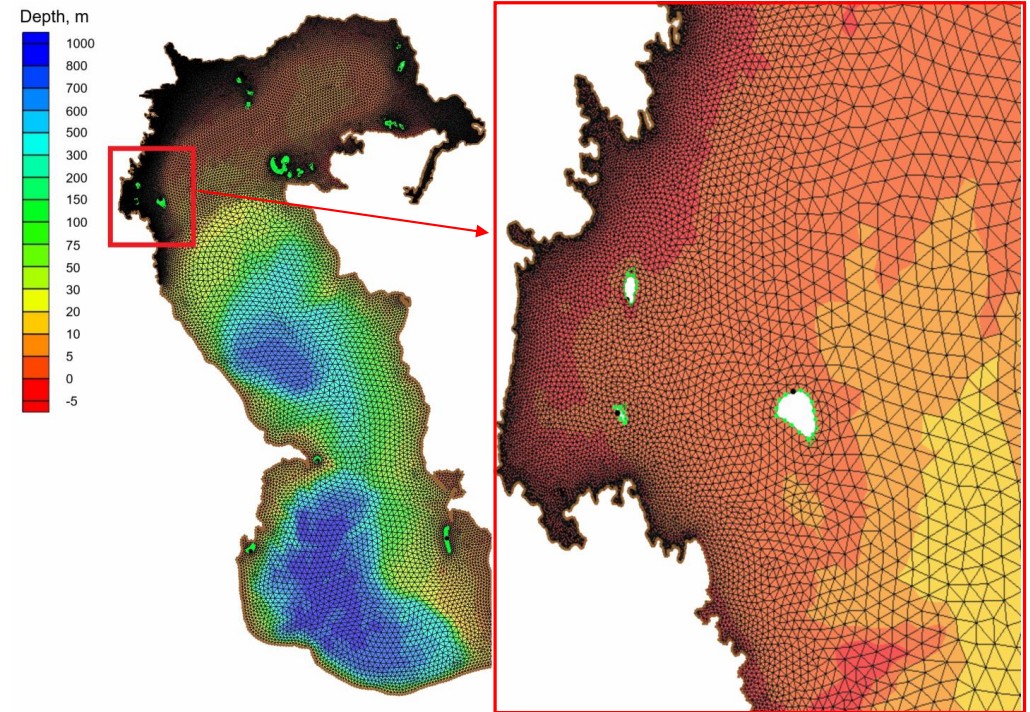


**Figure 1. The computational unstructured grid for ADCIRC model and the map of the Caspian Sea depth.**

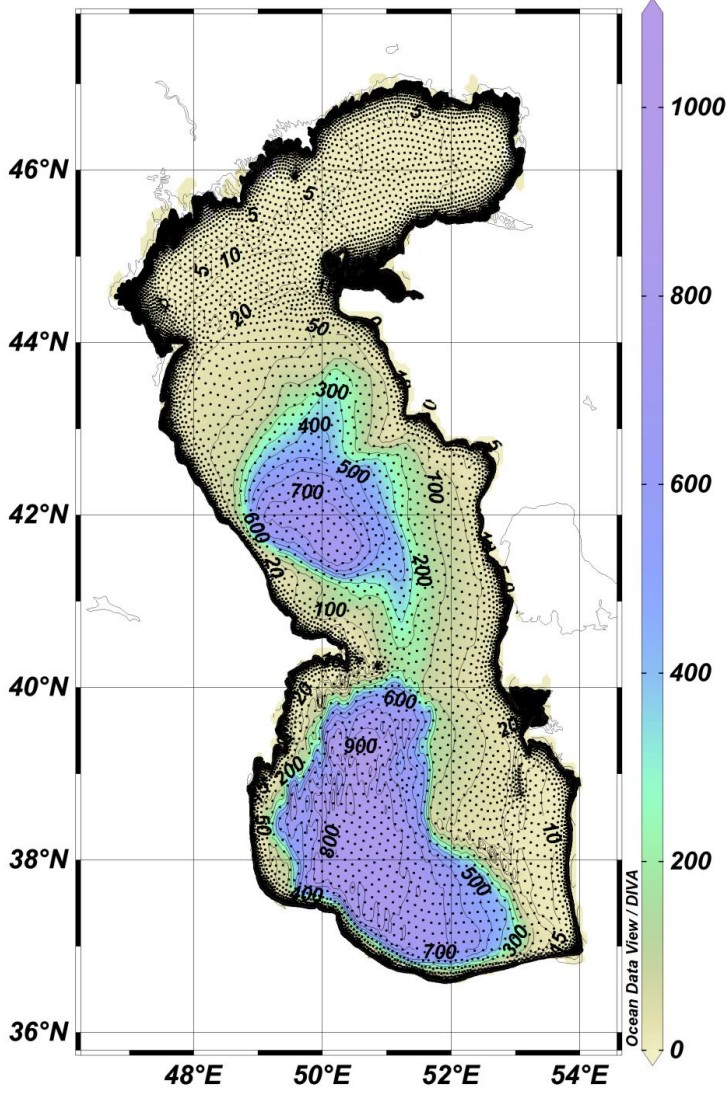

Figure 2. The computational unstructured grid for the WAVEWATCH III model and the map of the Caspian Sea depth.




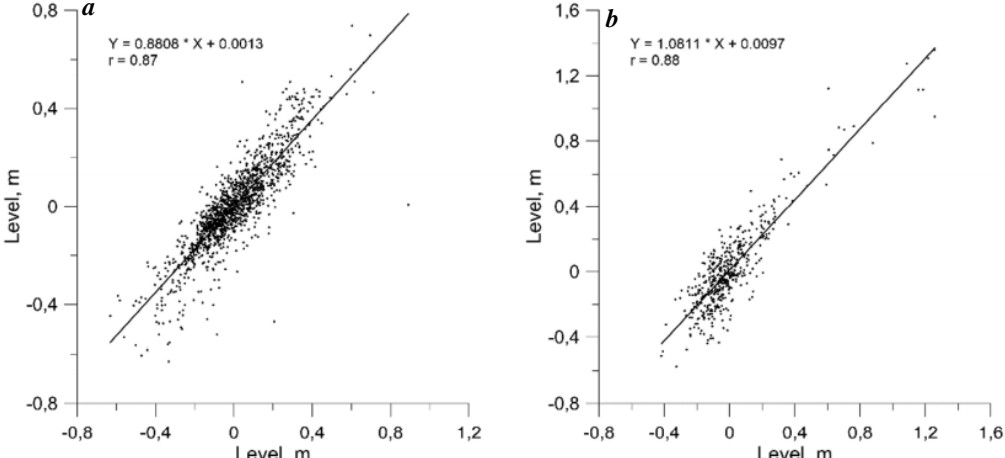


**Figure. 3. Sea level scatter diagram for 2009 (a) and 2015 (b) on the Tuleniy Island.**


**Figure 4. Sea level at Tuleniy station for surge 12-16 March 1995 (a) and 27 March to 1 April 2015 (b) (blue line – ADCIRC**
**+ SWAN, black – ADCIRC, red – measurement data).**

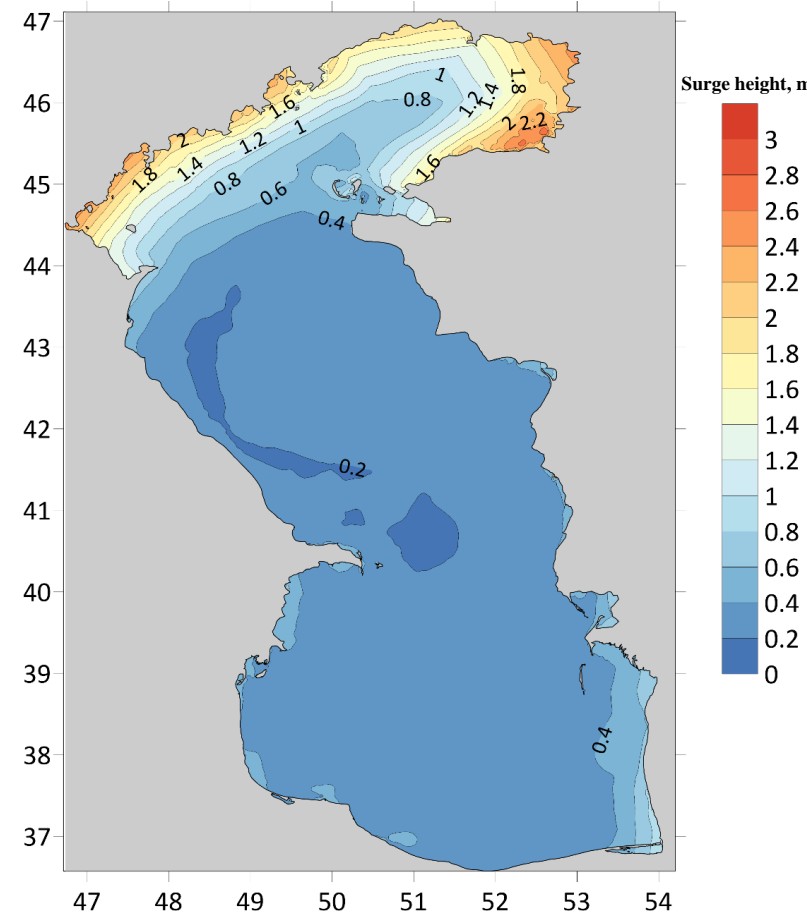

**Figure 5. Distribution of the maximum surge heights for period 1979 – 2017.**

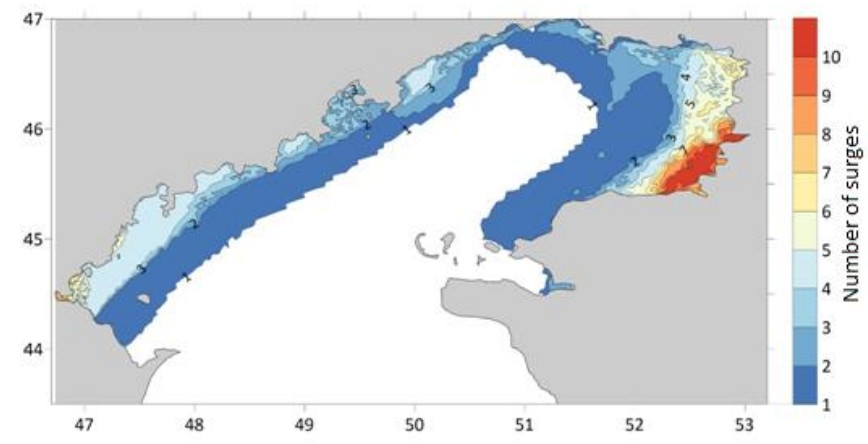

**Figure 6. Number of surges more than 1 meter in 1981.**


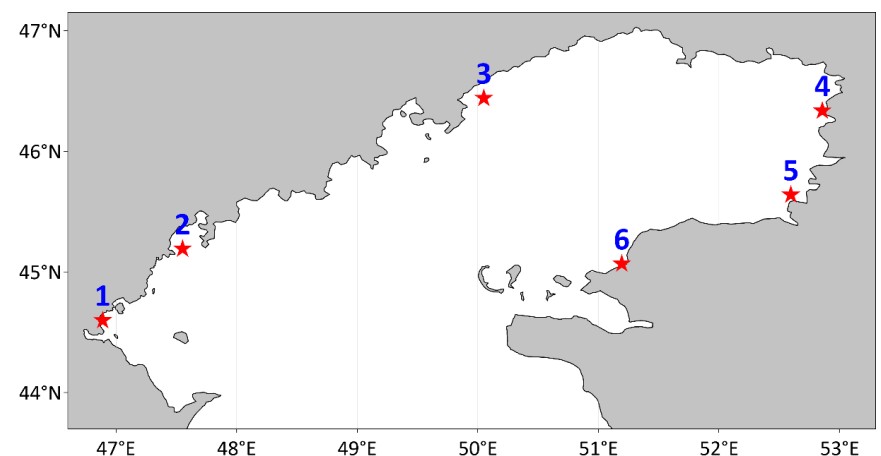

**Figure 7. Points for detailed analysis of number of surges in the northern part of the Caspian Sea.**

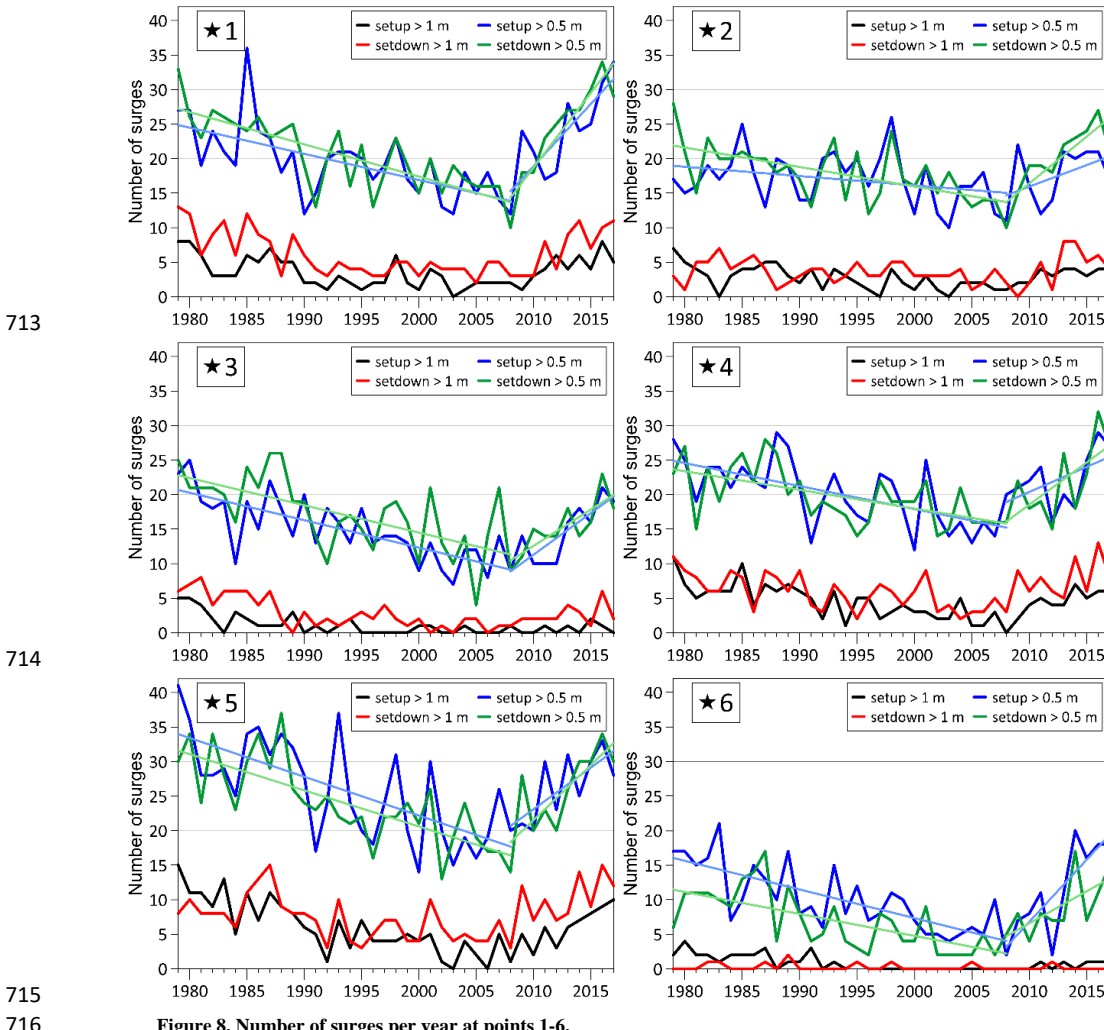

**Figure 8. Number of surges per year at points 1-6.**



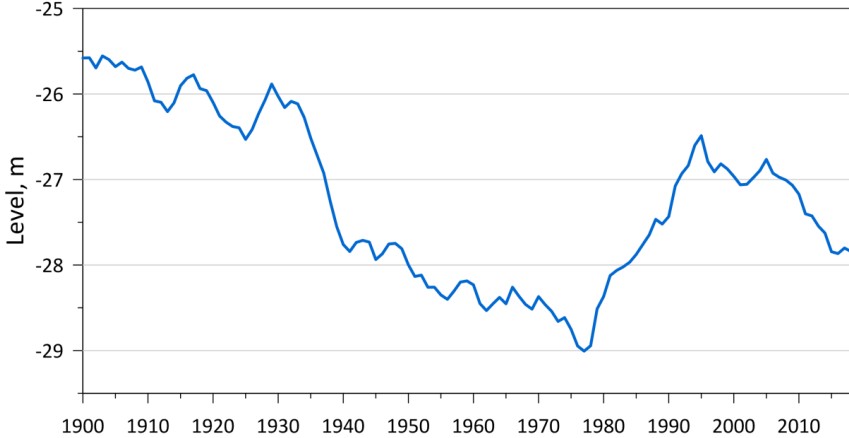

**Figure 9. The Caspian Sea mean annual level fluctuations for period 1900-2017.**

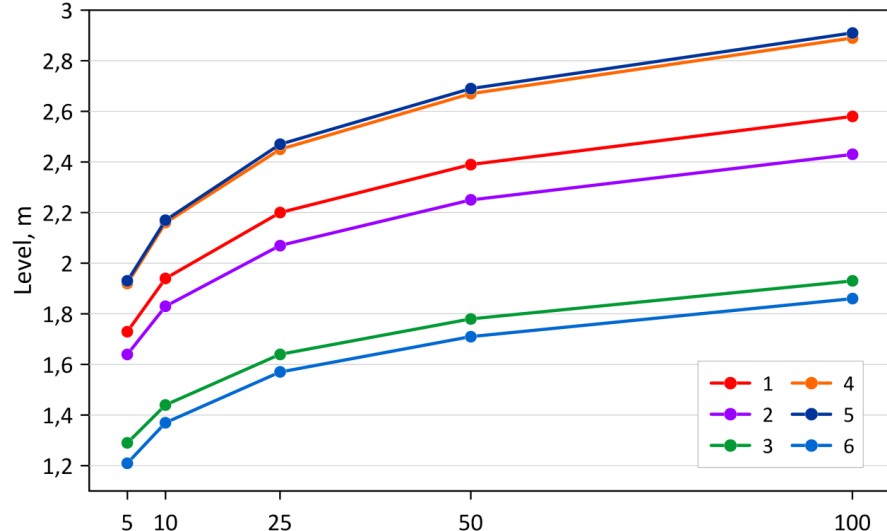

**Figure 10. Extreme sea level values at points 1–6 for the return periods 5, 10, 25, 50 and 100 years.**







**Figure 11. Extreme sea level values for the return periods 10 (a) and 100 (b) year**s.

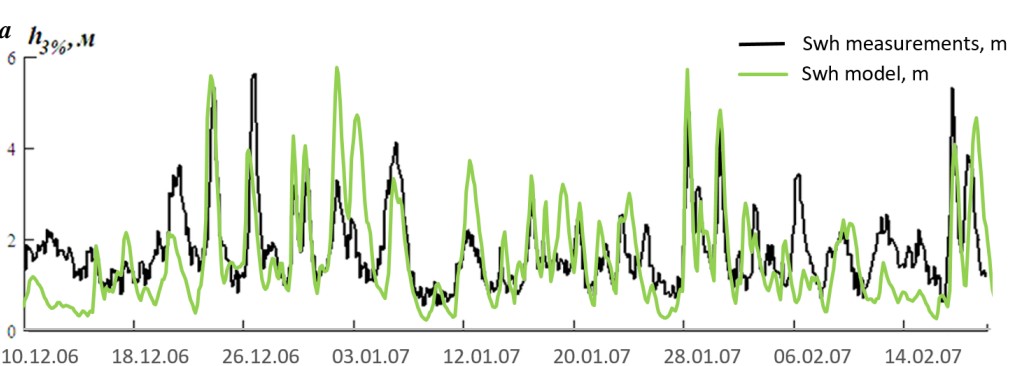


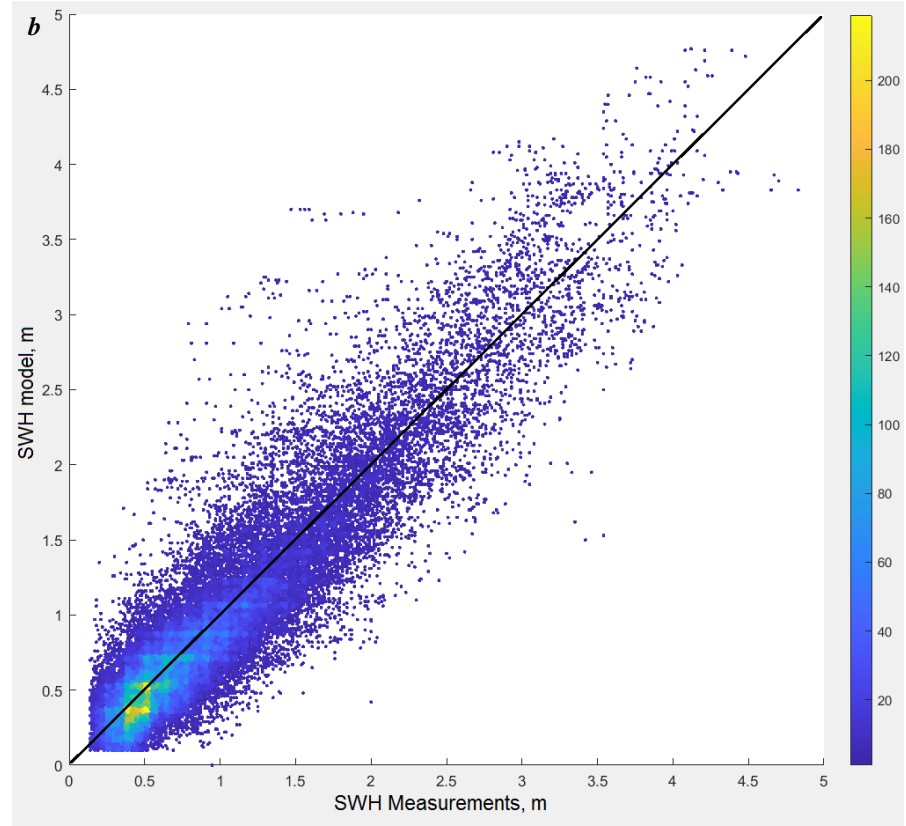


**Figure 12. The wave height of 3% probability of exceedance according to model results and direct measurements data in the point of 42°45.7' N, 49°41.6' E (a). Scatter plot of SWH derived from satellite and model data (b). N is the number of points per 0.05 × 0.05 m square.**



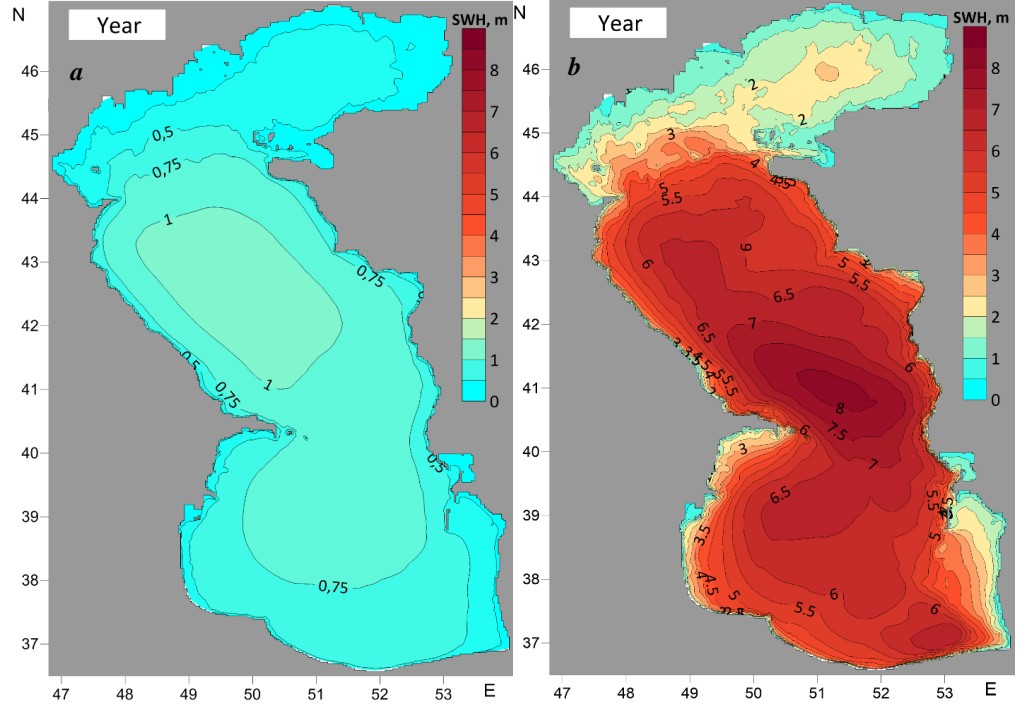

Figure 13. The long-term mean SWH (a), the maximum SWH (b) for period 1979 to 2020.




**Figure 14.** The long-term mean SWH (*a, c, e, g*) and maximum SWH (*b, d, f, h*) for different seasons.






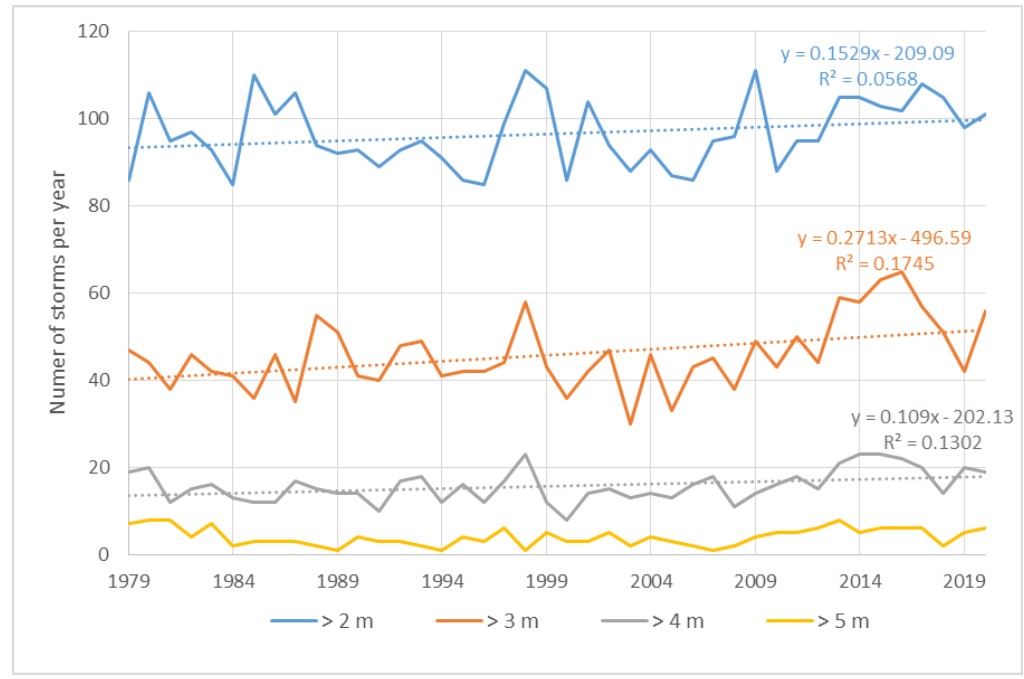

**Figure 15. The number of storms with different SWH thresholds per year and its linear trends for 1979 to 2020.**


Type I (19 %)

Type II (47 %)

Type III (34 %)

Pressure anomaly, hPa

**Figure 16. Anomalies of the surface pressure field, hPa, for each type of circulation from mean annual (1961-1990).**

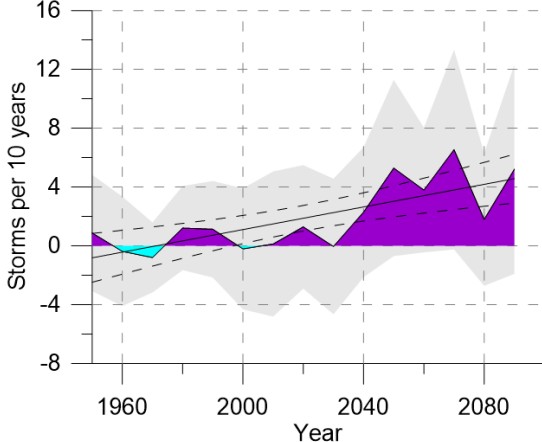



**Figure 17. Dynamics of anomaly per 10 years from the average for the period 1961-1990 the number of storm situations**
**for the ensemble of CMIP5 models according to the data of the Historical experiments (1950-2005) and RCP8.5 (2006-2100).  The**
**black line is the line trend and dotted black line its 95% confidence interval. Gray fill - intermodel spread of values.**

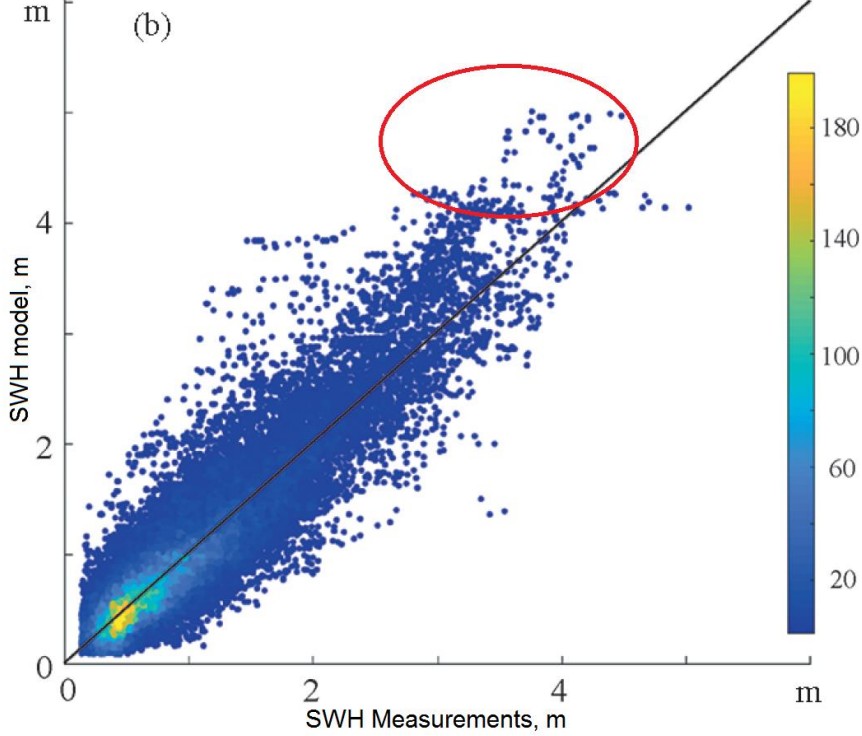


**Figure 18. Scatter plot of SWH derived from satellite and model data based on previous model configuration (Myslenkov**
**et al., 2018).**