# Peer review of "Storm surges and storm wind waves in the Caspian Sea in the present 1 and future climate 2 Anna Pavlova1,2, Stanislav Myslenkov1,2,3, Victor Arkhipkin1, Galina Surkova1 3 4 1Lomonosov Moscow State University, Faculty of Geography, Moscow, R"

_Natural Hazards and Earth System Sciences, 2021_

## Referee Comment (RC3)

**Referee comments on "Storm surges and storm wind waves in the Caspian Sea in the present and future climate"**

Anna Pavlova, Stanislav Myslenkov, Victor Arkhipkin, Galina Surkova

**General comments and recommendation:**

This paper aims at investigating storm surges and wind waves in the Caspian Sea over the last decades. It is based on a circulation model (ADCIRC) that can be coupled to a wave model (WAVEWATCH III and potentially another one named SWAN that is only briefly described). This article is scientifically well constructed and well presented. After the introduction, a description of the model and methods is included before the description of the results. More details could be included concerning the observational datasets used for model validation. The results and the discussion are sometimes merged in the same section. It could be clearer to separate the discussion from the results in another section. Interesting results are shown, with a quantification of the storm and wave intensities, their means, extreme values (including return periods) and potential trends. The results are innovative and merit publication. The manuscript should be rewritten with a strong attention to the grammatical form that limits the comprehension of the text. For more clarity, I would also recommend the use of tables to show the numbers directly included in the text, otherwise the text appears loaded with many numbers. The figure captions often lack information. Two other points need to be considered before publication: Figure 1 and 3 appear in a previous paper (Pavlova et al., 2020), with some text directly picked up from this publication. I would suggest summarizing the part common to this article already published. Second, the future prediction of storms based on CMIP5 only consider the storm changes related to changes of weather regimes frequency, in other words the dynamical changes related to atmospheric circulation. What about the thermodynamical changes impact on the storms? In other words, would it be possible to get stronger storms with the same frequency of weather type in a warmer climate? Maybe this question cannot be answered in this article, because it would require the use of a model chain using CMIP model output to force ADCIRC and WAVEWATCH in future projections, a work that has not been performed here if I understand well. Anyway, this should be discussed in the article.

After considering these general recommendations and the points presented below, a work that corresponds to major revisions, this article could be published in the journal.

**Point-by point list of comments:**

Abstract:

L. 12: SWH should be defined, for example in parenthesis after writing the corresponding words.

Introduction:

L. 33: maximum depth to -> maximum depth of ?

L. 38: please reformulate

L. 40 -> overlap **other** anthropogenic factors, knowing that the change of climate conditions is also attributable to anthropogenic factors.

L. 49: you could specify that circulation model reefer to ocean circulation model here, and not to atmospheric models.

L. 55: The paper (Lopatuhin et al., 2003) -> unusual formulation

L. 65: (Kudryavtseva et al., 2016) -> it is not needed to include this reference that is already used at the beginning of the sentence.

L. 67-68: this sentence should be reformulated.

L. 69-77: What is the limitation when using coarse gridded atmospheric data to force ocean and wave models? Is it possible to simulate wave with coarse gridded data, or is a downscaling approach is required beforhand? More discussion about the model resolution would be welcomed.

L. 81: "The papers (Rusu and Onea, 2013)" could be replaced simply by "Rusu and Onea (2013)".

Overall, the word « paper » is a bit familiar whereas the use of study or article is more common.

L. 93: « mean and extreme parameters » or « mean and extreme variables used to describe wind waves »?

Data and methods:

L. 104: the use [of] the unstructured grids

L. 139: GEN3 and KOMEN configurations with their parameters (cds, stpm, ect..) are not necessarily know by the reader. These should be explained in details, in the manuscript or in the supplement, or the text should be simplified excluding acronyms that cannot be understood by people that are not expert in these models.

Also how is managed the forcing by the coarse gridded atmospheric reanalysis that is applied to the non-regular ADCIRC grid, based on finer resolution?

L. 158 to169: the acronyms (e.g. ST6, IC0, JONSWAP, etc…) should be detailed or excluded. Another way to describe the model configurations would be to use a Table summarizing parameters, schemes, etc…

L. 208: to investigating -> to investigate

L. 225: For these days sea level pressure -> For these days, the sea level pressure

L. 251 to 257: a table could be used to show the model considered in the study. More information related to these models could be shown, including the resolution for example. Is a spatial interpolation on a common grid is applied before the clustering analysis?

Results and discussion

- Where are the weather stations Tuleniy Island and Makhachkala ? their location could be shown on a map or at least their geographical coordinates should be given. The station measurements sampled at these locations should be described.
- Why only the years 2009 and 2015 are considered, and only for one of the station? Why not showing a scatter plots including all the years? The two correlation coefficients shown in Figure 3 are 0.87 and 0.88, so we do not understand were do come from the 0.79 in the text. Scatter plots for the Makhachkala station could be also shown.
- Figure 1 and Figure 3 are already used in a previous publication (Pavlova et al., 2020), so this new article could be shortened with a reference to this previous article. Figure 1 might be reused because it shows the domain, but the evaluation shown in Figure 3 is exactly the same as the previous article, so the authors might consider excluding it from this new publication.
- How can we explain that the Figure 7 in Pavlova et al. (2020) show a 0.2 shift in the observation of the sea level as compared to those shown in Figure 4 of this new article?

L. 283: it could be interesting to highlight the location of the Volga river in Figure 5.

L 293: the sentence need to be reformulated.

L. 305: please, reformulate the sentence (grammatical construction)

L. 302 to 308: the sensivity experiments used to differentiate the contribution of wind and pressure changes to sea level change should be detailed (maybe with a table), including information like the experiment lengths (several years? Full period, one event?) as well as the complete protocol.

L. 321: 1 point -> point 1

L. 335: I would replace climate change by climate variability here, since this statement is verified both for long-term changes and variability at higher frequency.

L. 340: from 12 to 41 in the east (Fig.8) -> At point 6, the blue and green curves go below 5, isn't it?; « In the north, the values do not exceed 26. In case of surges of more than 1 meter, the maximum number of surges of up to 15 cases per year » -> It is not clear, which point are considered here?

L. 341: 1 point -> point 1. Please apply this change in the whole manuscript.

L. 343: the discussion on intra-annual variability should be based on numbers, included in the text or preferably included in a Figure or a Table.

Page 9: A table describing the mean numbers of surges of different magnitude (>0.5, >1, >2) for the different points, including variability, seasonal contrast and maximum/minimum values would be helpful in this manuscript.

Page 9: when including the Gumbel law for the distribution, the variable V should be defined as the parameters A and B. Same remark for "T" "m'" and « где N » that need to be explained when giving in the relationship between the period of the frequency and the number of values.

Page 9 and 10: The equations should be numbered as previously.

The equation considered to build the curves in Figure 10 should be mentioned in the text and maybe in the caption.

L. 372: on the west – 2.6 m -> on the west + 2.6 m, isn't it?

Figure 11: is the Gumbel law has been also used to build these maps?

L. 391: please reformulate the sentence.

L. 409-410: "In our results, we have a maximum value of SWH 8.2 m, which can be transformed to 10.8 m of 3% probability of exceedance. » -> How is get this result?

L435: « the significant linear trend » -> how is estimated the level of significance ?

L. 454: « Fig.16) » ->  (Fig.16)

L. 456-457: This statement is not clear.

L. 460:In he present climate -> In the present climate : I suppose that this reefer to 1961-1990 ? That's not clear.

L. 460-462 : this statement is not clear.

Conclusion:

L. 476-477: please reformulate the sentence.

L. 487: "The number of storms with the SWH ≥ 3 m » -> for more clarity, you could write "the annual number of storms…"

L. 507: SWH o about 8.9 m -> SWH of about 8.9 m

L 510: overestimates of the SWH.

Data availability: following the FAIR protocol, the data should be available on an open platform, without the need to contact the authors.

References: please homogenise the format of the references

Figures: all the Figures are well presented, but the figure captions should include more explicit details about the data and the methods used to build the curves and maps.